# Beyond Mechanical Tension: A Review of Resistance Exercise-Induced Lactate Responses & Muscle Hypertrophy

**DOI:** 10.3390/jfmk7040081

**Published:** 2022-10-04

**Authors:** Daniel Lawson, Christopher Vann, Brad J. Schoenfeld, Cody Haun

**Affiliations:** 1School of Kinesiology, Applied Health and Recreation, Oklahoma State University, Stillwater, OK 74078, USA; 2Duke Molecular Physiology Institute, Duke University School of Medicine, Duke University, Durham, NC 27701, USA; 3Department of Exercise Science and Recreation, Lehman College of CUNY, Bronx, NY 10468, USA; 4Fitomics, LLC, Alabaster, AL 35007, USA

**Keywords:** lactate, muscle hypertrophy, resistance exercise, resistance training

## Abstract

The present review aims to explore and discuss recent research relating to the lactate response to resistance training and the potential mechanisms by which lactate may contribute to skeletal muscle hypertrophy or help to prevent muscle atrophy. First, we will discuss foundational information pertaining to lactate including metabolism, measurement, shuttling, and potential (although seemingly elusive) mechanisms for hypertrophy. We will then provide a brief analysis of resistance training protocols and the associated lactate response. Lastly, we will discuss potential shortcomings, resistance training considerations, and future research directions regarding lactate’s role as a potential anabolic agent for skeletal muscle hypertrophy.

## 1. Introduction

History of Lactate

Lactate (La^−^) is the conjugate base of lactic acid (HLa), as described by Dr. L. Bruce Gladden [1,2], HLa is more than 99% dissociated into La^−^ anions and protons (H^+^) at physiological pH. This is a critical distinction and serves as a precept for this review. In 1780, a German-speaking Swedish apothecary and chemist, Carl Wilhelm Scheele, came to discover HLa in sour milk [3]. This new compound was called ‘‘Mjölksyra’’, translated from Swedish to English, which means ‘‘acid of milk’’ [3]. In addition to HLa, Scheele had also discovered chlorine, manganese, arsenic acid, barium, molybdenum, tungsten, fluorine, and numerous other compounds. Most notably, Scheele is best remembered for the modern discovery of oxygen (O_2_) [4]. Between the first discovery of HLa in 1780 and the muscle studies by Fletcher and Hopkins in 1907, a notably popular finding in 1808 by Berzelius was an elevated concentration of HLa in ‘the muscles of hunted stags’ and that its concentration was dependent upon how close the animal had been driven to exhaustion [5].

Ensuing research over the next century helped to elucidate further the role of La^−^, including several publications in 1859 by du Bois-Reymond stating that activity led to increased muscle acidity, relating this finding to that reported by Berzelius [6]. Around the same time, the German chemist and a principal founder of biochemistry, Justus von Lieber, demonstrated HLa was always present in dead tissue [7]. It took close to 100 years for chemists to delineate what acids really were, which, combined with the limitations of their technology, consequently led them to believe that they were measuring HLa as opposed to La^−^. La^−^ is a negatively charged ion, so once it was extracted from the blood, the molecule would combine with a proton to form what many know today as HLa. Unfortunately, this historical misunderstanding resulted in La^−^ being labeled as an agent of muscle fatigue or muscle “burn” [8].

The landmark study by Fletcher and Hopkins in 1907 demonstrated that at rest, La^−^ levels in amphibian muscle were low, whereas increased La^−^ was observed with an exercise stimulus [6]. They also demonstrated that La^−^ accumulates to significantly high levels during stimulation of muscles to fatigue and discovered that when the fatigued muscles are placed in an O_2_-rich environment, La^−^ subsequently disappears [6]. In 1930, Nobel laureate Otto Meyerhof also identified the accumulation of La^−^ and lack of O_2_ in exercising muscles, giving rise to the hypothesis that La^−^ was the result or indication of hypoxia [9].

The concept of “anaerobic threshold”—centered on inadequate O_2_ causing accumulation of La^−^—was introduced roughly 40 years later by Wasserman and McIlroy [10,11]. In 1964, Wasserman and McIlroy defined the “anaerobic threshold” as the O_2_ consumption (VO_2_) at which the respiratory exchange ratio (R) increased suddenly in a nonlinear fashion during the implementation of a progressive incremental exercise test [10]. A non-invasive procedure was subsequently developed to investigate the relationship between expired airflow (i.e., CO_2_ vs. O_2_) and heart rate (HR) across various time points to indicate when “lactic acidosis” occurs during incremental exercise. Other researchers would use a more direct approach to measure blood La^−^ at varying intensities during exercise, including muscle biopsies from the vastus lateralis (VL) and blood samples drawn from the earlobe or index finger [1,11,12,13,14,15,16].

The term “anaerobic threshold” implies La^−^ levels share an exclusive inverse relationship with O_2_ to the extent that La^−^ production is due to dysoxia. The term dysoxia has been defined as a low level of O_2_ availability, thus limiting cytochrome turnover and leading to O_2_-limited oxidative phosphorylation [17]. However, disagreements on terminology would ensue as researchers discovered that La^−^ was independent of O_2_, and a more neutral term “lactate threshold” would be adopted [18,19,20].

It is now well established that an increase in La^−^ levels can represent something other than low O_2_ availability and that hypoxic conditions are not the only cause for an accumulation of La^−^. In 1968, researchers Stansby and Jöbsis invented the surface fluorescence method for intramitochondrial nicotinamide adenine dinucleotides (NAD+/NADH) detection, using this technique on an isolated canine gastrocnemius-superficial digital flexor complex in situ. At the onset of muscle contractions, mitochondrial NADH became oxidized (NAD+) in conjunction with elevated La^−^ levels, suggesting sufficient O_2_ was present for oxidative phosphorylation [5,9,21]. Therefore, hypoxic stimulation of anaerobic metabolism via glycolysis did not explain the net La^−^ concentration. Additionally, several studies from Connett et al. (1983, 1984, 1986) [22,23,24] demonstrated that La^−^ could accumulate within slow-oxidative fibers (type I) during fully aerobic conditions without explaining O_2_-limited mitochondrial ATP production.

There is a long history of La^−^ and its role during endurance-type training. However, comparatively less research has examined La^−^ responses to resistance training and the role these responses play in the hypertrophic response. Exercise is known to acutely modulate numerous intracellular signaling pathways related to gene and protein expression, protein turnover, and myogenic activation of satellite cells (i.e., skeletal muscle stem cells). The last several decades have revealed potential mechanisms whereby changes in La^−^ production may influence these intracellular signaling pathways through metabolic and regulatory processes, as well as La^−^ acting as a signaling molecule [25,26,27,28]. Several metabolic, regulatory, and signaling pathways that La^−^ affects include cellular redox, reactive oxygen species (ROS), sirtuins, hydroxycarboxylic acid receptor 1 (HCAR-1), transforming growth factor-beta 2 (TGF-β2), and lactylation of histones. However, it is beyond the scope of this review to provide an in-depth review of each individual mechanism; for those interested in mechanistic actions, please refer to the recent review by Brooks and colleagues on the topic [29]. The next section (Section I) will provide an overview of La^−^ metabolism, how La^−^ has historically been measured, and describe processes related to La^−^ shuttling. Section 2 will explore mechanisms by which La^−^ may promote muscle hypertrophy or combat muscle atrophy. Section 3 will discuss La^−^ responses to resistance training that may be leveraged by resistance trainees to exploit these effects, and the review will conclude with a discussion and proposal of future research directions.

## 2. Section I

### 2.1. Lactate Metabolism

La^−^ is a metabolite produced as the end product of glycolysis, an anaerobic pathway rapidly producing adenosine triphosphate (ATP) without O_2_ involvement [30]. Briefly, glycolysis is a 10-step sequence of enzymatic reactions occurring in the cytoplasm of cells resulting in the generation of either two pyruvate or two La^−^ molecules. Notably, previous work suggests that La^−^ is almost always the end product of glycolysis [30,31,32]. During glycolysis, one glucose molecule results in two energy-rich ATP generated from two adenosine diphosphates (ADP) and two NADH generated from the reduction of NAD+ [33]. If the glycolytic pathway begins with a glycogen molecule, then three ATP are produced.

The conversion of pyruvate to La^−^—the final step of glycolysis—is catalyzed by the enzyme lactate dehydrogenase (LDH). Of relevance, there are at least 6 LDH isoenzymes or “isozymes” (e.g., LD1-6) that vary in structure, substrate affinity, temperature sensitivity, and tissue specificity [34]. This reaction also results in the reformation, or “recycling” of NAD+ so that glycolysis may continue.

It is possible for blood La^−^ levels to rise independently of O_2_ availability in exercising muscle cells. Various explanations exist, including (i) insufficient NADH shuttling to muscle mitochondria [5,21], (ii) variations in muscle fiber LDH isozyme content [35,36], (iii) increased muscle cell glycolytic flux due to increased levels of blood catecholamines [37], or (iv) a lack of uptake of blood La^−^ by other tissues (e.g., adjacent muscle fibers, heart, liver, brain) [38,39]. For example, predominantly fast-twitch fibers (e.g., muscle fibers expressing primarily myosin heavy chain [MHC] IIa or IIx proteins) have been shown to contain more LDH 5, which has a higher affinity for pyruvate allowing it to convert pyruvate to La^−^ more readily than LDH 1 [36]. Thus, more La^−^ may be generated and appear in the blood to a greater extent when fast-twitch fibers are recruited during intense exercise regardless of muscle O_2_ availability [36]. However, later studies would challenge the variation in the effects of LDH isozymes, concluding that the exact regulatory role of different LDH isozymes remains relatively unknown [40,41]. Furthermore, increased circulating levels of epinephrine and norepinephrine tend to occur at higher relative exercise intensities, and once bound to muscle receptors, an increase in extracellular calcium (Ca^2+^) and stimulation of cyclic adenosine monophosphate (cAMP) formation can occur, allowing for increased glycolytic flux regardless of muscle O_2_ availability [33]. Additionally, due to the increasing work rate from exercise, accelerated glycolysis may increase factors that activate glycogen phosphorylase and subsequently phosphofructokinase (PFK) including Ca^2+^, ADP, AMP, and decreased ATP. Mitochondrial density may also play a pivotal role in the lactate response as mitochondria prevent an increase in concentrations of inorganic phosphate (Pi) and ADP, both of which are activators of glycolytic enzyme activity and glycogenolytic flux [42]. When the aforementioned information is taken as a whole, La^−^ production can provide important insights for elucidating molecular underpinnings of resistance and endurance training adaptation.

### 2.2. Measuring Lactate in the Blood versus Muscle

Before considering the La^−^ responses to resistance training, it is important to understand how La^−^ is measured along with the reliability and validity of these measures. La^−^ analysis via blood sampling has become increasingly popular through the years because of the ease of sampling and improved accuracy of micro-assay methods [43,44,45]. Sampling blood La^−^ levels is an innocuous procedure that requires only a few drops of blood that can be easily obtained from the fingertip or earlobe using a lancet [46]. Other techniques such as isotope tracers and muscle biopsies may provide a more detailed analysis of La^−^ concentrations within working muscle during exercise, however these procedures are more invasive and thus less common in the literature [47,48,49,50,51,52,53]. Blood La^−^ levels are typically expressed in mmol/L (i.e., mmol, mM) whereas muscle La^−^ levels are expressed in mmol/kg dry weight, but can be converted to mmol/L intracellular water [54].

High intersubject variability for blood and muscle La^−^ measurements have been previously demonstrated during cycling exercise although there was a significant yet moderate intrasubject correlation between muscle and blood La^−^ (r = 0.71, *p* < 0.05) at an exercise intensity that elicited approximately 4 mmol/L of blood La^−^ with no statistical differences in concentration levels found between slow- or fast-twitch muscle fibers [16]. Another study found blood La^−^ rose linearly with muscle La^−^ during a 12-min cycling test at various intensities (30, 50, 70, and 90% of maximal O_2_ uptake [VO_2max_]), but had a clear leveling off between 4–5 mmol/min of blood La^−^ with a continual increase in muscle La^−^ [49]. Contrarily, Chwalbinska-Moneta et al. (1989) [55] found muscle and blood La^−^ concentrations to be highly correlated (r = 0.91) during a progressive cycling test, with an abrupt increase in La^−^ concentrations occurring around 51% of VO_2max_. In a different study comparing two different resistance-training protocols using the leg press exercise (5 sets × 10 reps @ 80% vs. 10 sets × 5 reps @ 80%) changes in muscle and blood La^−^ concentrations were found to be strongly correlated (*r* = 0.9), with significantly higher concentrations measured in the 5 sets × 10 reps protocol (*p* < 0.001–0.01) [48].

At rest, blood La^−^ levels may range anywhere between 0.3–2 mmol/L. Intense exercise lasting between 45 s to 2 min can result in levels up to 25 mmol/L, or 15–20 times above normal resting conditions [46]. Interestingly, resting blood La^−^ levels have also been demonstrated to increase by 47% during psychosocial stress independent of exercise [56].

It is important to consider the method by which La^−^ is measured since sampling sites and the time at which the sample is collected can potentially influence measurement accuracy. Blood La^−^ measurements from the fingertip and earlobe often show a strong correlation with one another, although the fingertip generally results in a higher blood La^−^ reading [57,58,59]. However, during prolonged steady-state exercise, this difference becomes smaller [60]. A study using three different sampling sites (fingertip, earlobe, and forearm vein) on three different modes of exercise (cycle ergometer, treadmill, and arm-crank ergometer) found a higher fingertip measurement as compared to the earlobe and forearm during the treadmill and cycle ergometer test. However, the arm-crank ergometer resulted in higher blood La^−^ measurements of the fingertip and forearm vein as compared to the earlobe [61]. These data might suggest that measurements taken more proximal to the site of the exercising muscle may result in higher blood La^−^ values which could also be a more accurate indication of muscle La^−^ concentration within exercising muscle at the time of measurement.

Significant differences in measured blood La^−^ have been reported when sampling capillary blood via finger stick, venous whole blood through venipuncture of the antecubital vein, and when plasma is separated from these methods; however, these samples have been shown to have a strong correlation [62]. One study has demonstrated venous blood La^−^ levels to be ~40% lower than plasma La^−^ concentrations (*p* < 0.001), finger capillary blood La^−^ ~8% higher than venous blood La^−^ (*p* < 0.01), and plasma La^−^ concentrations ~30% higher than that of finger capillary blood La^−^ concentrations (*p* < 0.01) [62]. The differences observed in La^−^ values based on the sampling site, such as between the fingertip and antecubital vein, suggest that La^−^ is likely to be taken up and metabolized by the surrounding musculature and other organs (i.e., heart and liver). Given these sampling differences, caution should be used when direct comparisons between different sites are made [62,63]. The redistribution of La^−^ to other tissues for intermediate metabolism and oxidation can be explained by various La^−^ shuttles, which are discussed in the next section.

### 2.3. The Lactate Shuttle Hypothesis

The cell-to-cell La^−^ shuttle provides a basic framework as to how La^−^ is metabolized [64] (Figure 1). La^−^ is first created within the cytosol of the cell, both at rest and during exercise. La^−^ can be thought of as a vehicle linking glycolytic and oxidative metabolism between cells that are La^−^ “producers” and La^−^ “consumers” throughout all tissues and organs. Glycolysis can occur in both cell types, although La^−^ concentration is greatest in highly glycolytic producer cells (i.e., type IIa and type IIx muscle fibers) and lowest in highly oxidative consumer cells (i.e., type I muscle fibers). Blood La^−^ concentration is defined as the difference between the accumulation of La^−^ into the blood and blood La^−^ removal. During exercise, specifically high-intensity training with short rest intervals, La^−^ is produced rapidly by the working muscles while La^−^ clearance is slowed. The result is an increased intramuscular La^−^ concentration and an increased output of La^−^ systemically. The mechanism through which La^−^ and the accompanying hydrogen (H+), move out of the contracting muscle is via monocarboxylate transporters (MCT). MCT1 and MCT4 are the primary transporters in skeletal muscle [65,66,67,68,69]. The use of a MCT allows La^−^ to move from the interstitial fluid, across the sarcolemmal membrane, into the blood and vice versa. Although skeletal muscle tissue appears to be one of the primary sources for La^−^ production and clearance during exercise, La^−^ that is not metabolized in the skeletal muscle may end up in circulation (e.g., plasma) where some of the La^−^ can be taken up via an MCT (e.g., MCT1) into red blood cells (RBC) as an intermediate step for transportation. Interestingly, La^−^ transportation into RBC may reduce the levels of La^−^ and H+ in the plasma, resulting in a larger gradient from interstitial fluid to plasma, which may positively influence the rate of release of these ions from exercising muscle tissue [70,71]. Once La^−^ has entered circulation, it can be transported to various tissues such as the liver, heart, inactive and active skeletal muscles, and other tissues, including the brain [72].

Once La^−^ has reentered consumer tissues, the substrate is then either further oxidized through gluconeogenic pathways, or it can be converted into pyruvate and then acetyl-CoA and aerobically metabolized via the citric acid cycle [73]. It has been proposed that due to the subsarcolemmal mitochondria being located near the arterioles, this likely influences the rate at that La^−^ is taken up from circulation and its subsequent oxidation [41]. Skeletal muscle is likely a major component of the La^−^ shuttle due to its large mass and metabolic capacity, as it not only produces large amounts of La^−^ during exercise but also uptakes and utilizes La^−^.

An intracellular La^−^ shuttle hypothesis has been used to help further explain the fate of La^−^ within skeletal muscle [35,74]. La^−^ is continuously produced within the cytosol, and the production rate increases relative to the glycolytic demands during exercise. An MCT (e.g., MCT1) shuttles La^−^ across the inner mitochondrial membrane, where a mitochondrial-specific LDH converts La^−^ back to pyruvate within the matrix of the mitochondria [35,75]. The pyruvate would then be oxidized to acetyl-CoA via pyruvate dehydrogenase (PDH), where it is then oxidized during the citric acid cycle to form NADH and FADH to enter the electron transport chain (ETC) and dispose of the H+. The final product of the ETC is the formation of ATP and water, with the water being formed via O_2_-accepting electrons (i.e., H+).

Another hypothesis for La^−^ metabolism is via an astrocyte-neuron La^−^ shuttle [76,77]. Glucose is the preferred energy source for neuronal energy metabolism [78]. Because of this, blood glucose may be more readily available to neurons and astrocytes via specific glucose transporters (GLUT3 for neurons and GLUT1 for astrocytes). As ATP use increases, glucose uptake via the neurons will also increase. Approximately 7% of cerebral energy requirements under resting conditions are met via La^−^ metabolism and up to 25% of the energy requirements being met during exercise when systemic La^−^ levels of 7 mmol/L are reached. The brain contributes substantially to systemic La^−^ production and utilization with approximately 13% and 8% under basal conditions. Under exercising conditions, the brain likely contributes up to 11% of the total La^−^ clearance [41]. It is important to conclude that skeletal muscle is the predominant tissue in La^−^ production and clearance during exercise; however, neurons and astrocytes play a key role in the uptake and oxidation of La^−^ during exercise as well. It is interesting to consider that La^−^ may play a supporting role in muscle hypertrophy via both neural (as an energetic substrate for motor neuron recruitment) and muscular mechanisms (discussed in detail in the following section).

## 3. Section II

La^−^ functions as an intracellular metabolite and extracellular ligand that mediates cell signaling via autocrine, paracrine, and endocrine functions. In this section, we will discuss the potential mechanisms La^−^ may contribute to exercise-induced skeletal muscle hypertrophy.

### 3.1. Lactate-Stimulated Testosterone Production

Increases in blood La^−^ levels have been demonstrated to accompany incremental increases in testosterone. Following brief intense exercise, testosterone is mainly produced and released from the interstitial or Leydig cells of the testes [79]. Research demonstrates that testosterone can increase independently of luteinizing hormone (LH) in plasma following exercise [80,81,82]. Lactate-stimulated increases in intracellular cAMP levels may influence testosterone production in Leydig cells [82]. One study demonstrated lactate-stimulated testosterone production in rat Leydig cells that were incubated with 10 mmol/L of La^−^ [83]. Lu et al. (1997) found a dose-dependent increase in testosterone levels and cAMP production following a 60-min incubation period of testicular tissue from rats in 0.1–10 mmol/L of La^−^. These results suggest La^−^ has a stimulatory effect on secreting testosterone via increased testicular cAMP levels. Testosterone is considered an anabolic hormone, playing a primary role in activating the mammalian target of rapamycin (mTOR), a vital regulator of muscle protein synthesis [MPS] [84,85]. It should be noted that acute endogenously elevated levels of testosterone following resistance training have not been demonstrated to meaningfully influence muscle hypertrophy [86,87,88,89,90]. To our knowledge, lactate-induced testosterone increase in humans has not been previously documented, and caution is warranted in making strong inferences. Nevertheless, evidence from multiple studies demonstrates when initially low testosterone levels are increased within the physiological range via exogenous testosterone treatment, fat-free mass and muscle volume tend to display measurable increases over the course of weeks [91,92,93,94,95,96]. Resistance training-induced increases in La^−^ levels and subsequent increases in testicular cAMP levels might enhance testosterone production, which may help to support muscle mass. However, there is insufficient direct evidence of this phenomenon in humans. Moreover, whether the magnitude of such increases, if they do indeed occur, would meaningfully enhance anabolism remains speculative.

### 3.2. Lactate-Related Epigenetic Modification

Epigenetic modifications are generally described as alterations in gene expression not attributable to changes in DNA sequence [97]. The three main types of epigenetic mechanisms are: (i) non-coding RNA (ncRNA)-associated gene silencing [97,98,99]; (ii) DNA methylation; and (iii) histone modification, which will be discussed below. Briefly, DNA is bound to proteins called histones which function as structural support. This DNA-protein complex is composed of four histones (H2A, H2B, H3, and H4) and is termed chromatin [100]. Post-translational modifications such as histone acetylation and lactylation occur on the protruding-tail of the histone (2). Histone acetylation occurs on lysine residues via the addition of acetyl groups by histone acetyltransferase (HAT); resulting in a weakened histone-DNA interaction, thus making genes more accessible for transcription factors [101,102]. Conversely, the removal of acetyl groups from lysine residues (post-transcriptional modification) is done via histone deacetylases (HDACs) [103], resulting in the suppression of general gene transcription [104]. Previous in vitro work using HCT116 cells has shown that a La^−^ concentration of 30 mM inhibits HDAC activity by ~20%; yielding results similar to other well-established HDAC inhibitors (i.e., butyrate and trichostatin A) [105]. With high-intensity exercise, muscle La^−^ can reach concentrations in excess of 60 mM [106] thus, it is conceivable that La^−^ can acutely modulate acetylation signaling cascades resulting in increased gene expression [107]. Interestingly, in 2019, Zhang and colleagues proposed a novel histone modification-termed lactylation-identified in HeLa cells and bone marrow-derived macrophages (BMDMs)-which acts similarly to acetylation of lysine residues [108]. Of interest, the findings of this study suggest that lysine lactylation may be regulated through glycolysis, and when LDH is genetically deleted, lactylation of lysine residues are ablated [108]. To date, in the context of skeletal muscle, the role of La^−^ on histone modification, as well as DNA methylation and nc-RNA associated gene silencing, remains unclear thus, it is an area ripe for future research.

### 3.3. Anabolic Effects of Lactate

It is now established that La^−^ plays a critical role not only in mediating specific metabolic adaptations but may also promote a hypertrophic response in skeletal muscle tissue. However, the exact mechanism(s) has not been well elucidated yet in the literature.

One potential mechanism La^−^ may influence is muscle cell myogenesis [109,110]. In short, myogenesis is the process through which new muscle cells are created [111]. This process is known to occur in embryo cells (i.e., embryogenesis). Adult myogenesis involves the process through which skeletal muscle-specific satellite cells (SCs) are activated, thus ending their quiescence state and entering the cell cycle. SCs then differentiate into myoblasts, which can fuse to injured myofibers or may even potentially assist in developing new myofibers to remodel muscle tissue [112]. This process of regeneration is tightly modulated via the relationship of key transcriptional regulators such as paired box 7 (Pax7) and the basic helix-loop-helix family of myogenic regulatory factors (MRF), which include myogenic factor 5 (Myf5), myogenic determination protein (MyoD), myogenin, and MRF4 [113,114]. Pax7 is expressed in SCs during their quiescence state [115] and may have a regulatory role of MyoD [116]. Once SCs are activated, Myf5 is expressed, followed by MyoD, myogenin, and MRF4, entering terminal differentiation into new or previously existing fibers [115]. Specific expressions of structural and enzymatic proteins, such as myosin heavy chains (MHC), are achieved during terminal differentiation. It’s important to note that the process of creating new muscle cells (i.e., hyperplasia) has been demonstrated in vitro and avian models, but it remains questionable whether or not this process occurs in humans during the performance of traditional resistance training protocols [117]. It is believed that in adult human skeletal muscle, SCs fuse to existing fibers, donate their nucleus, and allow for an expanded myonuclear domain of an existing muscle cell [118]. This inherent ability of SCs to donate additional nuclei may provide a unique capability to potentiate skeletal muscle repair, maintenance, and hypertrophy via increased transcriptional capacity for protein, thus potentially playing a major role in muscular adaptation in response to physical training [118,119]. However, it should be considered that increases in myonuclei per muscle fiber may not always be the end result of a training intervention, as the theory of myonuclear turnover would suggest a process whereby nuclei are eliminated and replaced with a new nuclei to support remodeling of muscle tissue [120]. A direct mechanism through which La^−^ mediates myogenesis and SC activity remains unclear. It has been suggested in vitro that elevated La^−^ (10 mM) may modulate myogenesis via increased MyoD expression [121]. Other studies have demonstrated increases in myotube diameter and nuclei per fiber in both undifferentiated and differentiated C2C12 myoblasts incubated in a medium containing La^−^ [109,110]. Evidence also La^−^ regulates myogensis via mitogen-activated protein kinase (MAPK) pathways by activating extracellular signal-related kinase-1/2 (ERK1/2) [122] and decreasing p38 MAPK [123]. La^−^ may signal an increase of follistatin [Fst] [110], a potent antagonist to myostatin (MSTN)—a known inhibitor of muscle growth—both of which are key regulators in myogenesis [124]. It should be considered that the discussed mechanism(s) may not have an immediate benefit to muscle growth; rather, frequent exposure to La^−^ chronically may activate and/or potentiate SC activity resulting in enhanced skeletal muscle hypertrophy over time.

Mechanical load/stress, intracellular Ca++, hypoxia, and redox state has been proposed as training-induced stimuli that may potentially regulate skeletal muscle hypertrophy through various signal transduction pathways [125]. Skeletal muscle mass is the result of muscle protein synthesis (MPS) and muscle protein degradation (MPD), both pathways dictating a positive or negative hypertrophic environment for skeletal muscle hypertrophy [126,127]. Here, we propose La^−^ may also have a potential role in regulating skeletal muscle hypertrophy (Figure 2). Important to the discussion of La^−^ and training-related skeletal muscle hypertrophy is protein kinase B (Akt)/mammalian target of rapamycin (mTOR) pathways, p70S6K and 4EBP1 (downstream targets of mTOR), as well as the mitogen-activated protein kinase kinase (MAPK)/ERK1/2 pathway. Studies have demonstrated in rats when La^−^ is either injected or orally administered, an upregulation in genes related to MPS (insulin-like growth factor 1 receptor [IGF-1], Akt, mTOR) was observed in addition to phosphorylation of mTOR [128,129]; evidence suggesting La^−^ may promote skeletal muscle hypertrophy via increased MPS. The MAPK/ERK pathway links communication from a receptor on the surface of the sarcolemma to the DNA within the cell’s nucleus. ERK1/2 activates P90 ribosomal S6 kinase (i.e., p90RSK) as a result of mechanical tension, as well as stimulating SC proliferation and differentiation [130,131]. The MAPK/ERK pathway may be mediated via the lactate-specific G protein-coupled receptor 81 (GPR81), also known as hydroxycarboxylic acid receptor 1 (HCAR-1). La^−^ has been previously reported to activate the ERK1/2 pathway in L6 cultured skeletal muscle cells via GPR81 [132]. Furthermore, of interest, La^−^ may activate calcineurin [26], a Ca++/calmodulin-dependent serine/threonine protein phosphatase, which could influence the hypertrophic response in slow-twitch muscles (i.e., soleus) [133,134].

A study by Willkomm et al. (2014) demonstrated in C2C12 cells infused with 2-h intermittent La^−^ (10 mmol/L and 20 mmol/L) or a lactate-free solvent (control group) over a 5-day period (10 days for cells treated with 20 mmol/L of La^−^). Intermittent La^−^ exposure triggered proliferating SCs to withdraw from the cell-cycle and potentiate early differentiation (e.g., increases in the Pax7 and Myf5) independent of the La^−^ concentration (10 mmol/L vs. 20 mmol/L) but delayed late differentiation in a dose-dependent manner (e.g., reduced expression of myogenin and MHC compared to the control group). On Day 2 and Day 3, 10 mmol/L measured similar content of myogenin- and MHC-positive nuclei compared to the control group, whereas 20 mmol/L differentiated to the same extent as 10 mmol/L and the control group by Day 10. Markers of oxidative stress were also increased as a result of La^−^, which was found to mostly affect the areas around the nuclei where mitochondrial density is greatest. Interestingly, these effects were reversed by adding antioxidants (L-ascorbic acid, N-Acetyl-L-cysteine, and linolenic acid), suggesting that the lactate-induced changes are mediated to some degree via ROS. However, the lactate-ROS relationship remains unclear, with conflicting findings in the literature. An intriguing finding presented in the study was intermittent and acute La^−^ exposure, representative of a training microcycle, initiated SCs to withdraw from the cell-cycle and into early differentiation, however, continued exposure hindered late differentiation.

An in vitro study by Oishi et al. (2015) incubated C2C12 cells with 10 mmol/L of La^−^ or 10 mmol/L of La^−^ plus 5 mmol/L caffeine (LC). Significant increases in myogenin, Fst protein levels, and phosphorylation of p70S6K were measured in addition to decreased Mstn levels relative to the control group incubated in 5 mmol/L of caffeine only. The LC group furthered the anabolic effects versus La^−^ alone by increasing protein levels of Pax7, MyoD, and Ki67 in addition to myogenin. This suggests that La^−^ potential hypertrophic effects may be potentiated by caffeine. However, it is important to consider that the cells were incubated for 6 h to achieve these adaptations. It’s unlikely that this timeframe could be achieved or even sustained during normal exercise conditions in humans, thus calling into question the practical implications of the findings.

In an in vivo study by the same researchers [110], mice were either orally administered daily with LC (lactate: 1000 mg/kg; caffeine: 36 mg/kg) or provided volume-matched sterile water in exercise-only and sedentary groups. The exercise was conducted on a treadmill for 30 min at a low intensity every other day for 4 weeks. Resting blood La^−^ levels for all groups were 2.38 ± 0.18 mmol/L, which changed to 1.93 ± 0.16 mmol/L in exercise-only and up to 7.85 ± 2.63 mmol/L in the LC group. The LC and exercise-only significantly increased muscle mass of the tibialis anterior (TA) and the gastrocnemius (GA), with the LC group demonstrating the greatest increases over the exercise-only and sedentary control groups. However, myofibrillar protein concentration remained unchanged between the two groups. There was no indication that the exercise-only group had increased SCs activity, whereas the LC group exhibited significantly increased myogenin expression, indicating increased SCs activity. In addition to these findings, the LC group demonstrated significant increases in Fst expression relative to both the exercise-only and the sedentary group, with decreased myostatin in the TA of the LC group only. These findings suggest that low-intensity aerobic exercise coupled with La^−^ and caffeine may signal increases in SC activity and Fst.

Willkomm et al. (2017) conducted two experiments incubating C2C12 cells in 20 mmol/L of La^−^ or lactate-free solvents (purposed as control groups); a differentiation medium (DM) was used for all C2C12 cells. In the first experiment, myoblasts were treated for time intervals of 15 min, 30 min, 1 h, 4 h, or 24 h; samples were collected at each the end of the designated time intervals and then lysed for Western blotting (WB). Additional myoblasts were treated for 15 min, 30 min, 1 h, and 4 h and then analyzed via immunofluorescence (IF). WB of samples collected at 30 min and 1 h of treatment revealed p38 MAPK phosphorylated in the control group but not 20 mmol/L of La^−^. WB also revealed that H3K4me3, a key player in myogenesis [123], decreased with the La^−^ treatment and was no longer detectable after 24 h. When compared to the control group (0 mmol/L), gene analysis reported La^−^ decreased Myf5 mRNA (*p* = 0.033) while preventing expression of myogenin (*p* = 0.046) and MHC1 (*p* = 0.044). La^−^ seemingly decreased MHC2 expression but was not deemed statistically different. The data from this experiment suggests La^−^ regulates myogenesis via decreased both p38 MAPK and H3Kme3, which could influence skeletal muscle regeneration and hypertrophy [135,136]. The second experiment treated myoblasts for 2 h a day over 5 days to replicate a realistic training environment. To determine if the effects of La^−^ are regulated via p38 MAPK inhibition, some myoblasts were treated with a lactate-free solvent containing SB203580 (10 μM), a specific p38 MAPK inhibitor. IF analysis revealed an abundance of Pax7- and Myf5-positive nuclei from La^−^ and SB203580 treated myoblasts compared to those treated in solvents free of La^−^ and SB203580. Contrarily, WB and IF analysis revealed myogenin and MHC, indicators of end-terminal differentiation, were significantly reduced as a result of La^−^ and SB203580. Collectively, this study suggests a mechanism whereby La^−^ may actually reduce SC differention which consequently could reduce adaptations favorable for skeletal muscle hypertrophy.

In addition to the in vitro data, the authors [123] conducted additional analyses of human tissue from subjects randomly assigned to a moderate intensity (STD), high intensity (HIT), or maximal eccentric (ECC) resistance training protocol (STD = 3 sets × 10 unilateral leg extensions at 75% of 1RM with 3-min rest periods; HIT = 1 set × 20 unilateral leg extensions to failure without rest; ECC = 3 sets × 8 unilateral leg extensions at 100% maximum eccentric force with 3-min rest periods; blood La^−^ data was not provided for ECC and therefore ECC will not be included in this discussion). A full breakdown of the training protocol can be found here [137]. The HIT protocol measured the highest blood La^−^ measurements across 2, 4, 6, 8, and 10 min (HIT vs. STD; 5.92 vs. 3.44, 6.04 vs. 3.23, 5.89 vs. 2.92, 5.47 vs. 2.59, 4.83 vs. 2.33 mmol/L). Both p38 MAPK and H3K4me3 decreased following the HIT protocol, which the authors speculate was partially attributable to the higher blood La^−^ levels measured in the HIT group. However, this is speculative and without muscle La^−^ data it is even more difficult to assess if the blood La^−^ levels are representative of the intramuscular concentrations. While the data presented indicate higher levels of La^−^ may decrease p38 MAPK and H3K4me3, it is important to consider that the reduced signaling (i.e., p38 MAPK and H3K4me3) may be transitory in order to upregulate juxtaposing pathways for adaptation.

Tsukamoto et al. (2018) treated cultured C2C12 cells with 10 mmol/L of La^−^ for 5 days to investigate whether La^−^ influences myogenic differentiation. Myoblast fusion was accelerated due to La^−^ over the 5 days compared to the control group treated with sodium chloride. MyoD expression was also significantly enhanced on Day 3 and Day 5, whereas no changes were observed in Myf5 or myogenin. The myotubes differentiated with 10 mmol/L of La^−^ also exhibited significantly more MHC proteins. This was explained via increased expression of Myh4 (encoding for MHC type IIb). No changes were observed in p70S6K, which is in contrast to the findings by Oishi et al. (2015). The researchers then treated the cells with various concentrations of La^−^ (0 mmol/L, 2 mmol/L, 4 mmol/L, 6 mmol/L, 8 mmol/L, and 10 mmol/L). They determined that expression of both Myh4 and Myh1 (encoding for MHC type IId/x) mRNA expression was dose-dependent, as higher levels of La^−^ (i.e., 6 mmol/L, 8 mmol/L, and 10 mmol/L) exhibited greater increases in the expression of these genes.

An in vivo study by Tsukamoto et al. (2018) also investigated La^−^ hypertrophic and regenerative abilities. Mice were first injected with glycerol to induce necrosis and then injected with 500 mg/kg of La^−^ solute daily for 28 days. Individual fibers were measured on Day 7, Day 14, and Day 28 to assess relevant changes. Gene expression analysis demonstrated significant increases in Myh4 within the lactate-treated group, as well as trends for increased expression of MyoD in TA muscles. Both groups’ necrotic fibers were nearly completely regenerated by Day 28 following the injury. However, individual fiber assessment indicated that the lactate-treated group experienced significantly greater hypertrophy by Day 28 than the control group (*p* < 0.001). Similar results were found by Ohno et al. (2019) when orally administered La^−^ followed a cardio-toxin injection into the TA of mice. They demonstrated significant increases in muscle weight and fiber cross-sectional area, as well as increases in Pax7 in the TA, indicating increased activity of satellite cells. When lactate was administered into the culture medium of C2C12 cells during myotube formation, significant increases in protein content, fiber diameter, length, and myonuclei were found as compared to the control group.

Ohno et al. (2018) treated C2C12 cells with sodium La^−^ concentrations of 0, 5, 10, and 20 mmol/L. Following the administration of La^−^, changes in the relative phosphorylation levels of EK1/2, ERK1/2, 90RSK, Akt, mTOR, p70S6K, FoxO3a, ULK1 as well as relative expression levels of LC3B-II to LC3B-I were assessed only for the 20 mmol/L conditions at 2 h. La^−^ administration (20 mmol/L) significantly increased the expression levels of phosphorylated MEK1/2 (p-MEK1/2), p-ERK1/2, and p-p90RSK (*p* < 0.05). There was no significant interaction in the expression levels of Akt, mTOR, p70S6K, p-FoxO3a, p-ULK1, and LC3B-II. Myotube diameter increased the most with 20 mmol/L of La^−^ relative to the control (*p* < 0.05). An interesting mechanism investigated in this study was the lactate-GPR81 relationship, with the authors reporting greater expression of GPR81 in myotubes relative to myoblasts (*p* < 0.05). Some C2C12 cells were incubated with only 3,5-DHBA, a specific agonist for GPR81, at various concentrations over the same time period as the La^−^ treatment. The treatment of 3,5-DHBA was shown to significantly upregulate the relative phosphorylated levels of MEK1/2, ERK1/2 and p90RSK (*p* < 0.05) similarly to the La^−^ treatment. Additional cells were incubated with La^−^ and/or the MEK inhibitor, U0126. Both p-ERK1/2 and p-p90RSK were significantly lower in cells treated with La^−^ and U0126 compared to La^−^ treatment alone (*p* < 0.05). La^−^ and U0126 treatment consequently resulted in no change in myotube size compared to La^−^ alone (*p* < 0.05). These findings suggest that a lactate-induced hypertrophic response may be mediated via the La^−^ receptor GPR81, which activates the MEK/ERK pathway. There are limitations to this study. The reported increase in myotube diameter was a result of continuous extracellular La^−^ exposure for 5 days with 20 mmol/L being the only condition that was statistically significant relative to the control (*p* < 0.05). Achieving this level of La^−^ in the blood or muscle tissue during exercise would be difficult, and sustaining such a level close to the timeframe in this study is likely physiologically impossible. At 2 h, 20 mmol/L of La^−^ induced a meaningful response in various anabolic pathways. This timeframe is likely more reasonable as it relates to a training session, however, it’s still likely impossible to sustain considering 8 mmol/L is the average cutoff point measured during a VO_2_Max test typically lasting 10–20 min [138]. It would have been insightful and more practically meaningful had the researchers also investigated the response of the aforementioned variables in the 5 and 10 mmol/L treatments, as these are likely more sustainable levels during prolonged exercise such as resistance training.

Cerda-Kohler et al. (2018) intraperitoneally injected mice with La^−^ or volume-matched phosphate-buffered saline to investigate the potential intracellular signaling pathways of skeletal muscle. Tissue samples from the quadricep, soleus, and EDL were harvested 40 min following the injection of La^−^ or saline. Blood La^−^ levels rose quickly following the injection of La^−^ but not the saline, with no changes in blood glucose or insulin levels for either condition. Anabolic signaling following the La^−^ administration increased Akt in the quad (2-fold; *p* = 0.02) and EDL (2.2-fold; *p* = 0.02), ERK1/2 (3.5-fold; *p* = 0.004) and p70S6K (1.9-fold; *p* = 0.01) in quadriceps, S6 in both the quadriceps (6.3-fold; *p* = 0.01) and EDL (2.3-fold; *p* = 0.01), with no changes in the soleus. Interestingly, AMPK displayed a tendency to increase (1.7-fold; *p* = 0.08) coupled with a significant increase in ACC (1.5-fold; *p* = 0.04), which are both key indicators for oxidative metabolism. Although the exact blood La^−^ levels were not provided, an interesting observation was that lactate increased up to ~20 mmol/L at 5 min and remained elevated above ~10 mmol/L at 30 min. These are the same levels that previous studies reported anabolic signaling from La^−^ concentrations ranging from 10–20 mmol/L for an extended duration of 2 or more hours. What makes these findings intriguing is that one bolus of intraperitoneally injected La^−^ in mice was able to significantly increase anabolic signaling pathways within 40 min. However, in humans achieving La^−^ levels between 10–20 mmol/L could prove challenging, especially obtaining values closer to 20 mmol/L. It is unclear if one large influx of La^−^ is sufficient to signal anabolic pathways in muscle or if prolonged continuous exposure to this metabolite is required. This must be considered as the blood La^−^ levels remained elevated to a significant degree in the mice, however, in humans, lactate clearance following the cessation exercise occurs rather quickly reaching baseline values within 30 min in humans [139].

A follow-up study by Ohno et al. (2019) demonstrated that orally administered La^−^ in mice 5 days a week for 2 weeks increased TA weight and fiber CSA relative to the control group. When mice were injected with a cardio-toxin injection (CTX), La^−^ appeared to support muscle regeneration by increasing TA muscle weight (24% absolute and 29% relative) with increases in fiber CSA of 44% at 2 weeks relative to the week one measurement. Additionally, Pax7-positive nuclei significantly increased as a result of orally administered La^−^ compared to the control group following CTX injection (0.08/myofiber vs. 0.02/myofiber; *p* < 0.05). C2C12 cells were cultured in 20 mmol/L of La^−^ for 5 days to investigate the effects of La^−^ on myotube formation. After 5 days, it was reported that myotubes increased in diameter, length, and myonuclei compared to the control (*p* < 0.05). The limitations are again the duration of exposure to La^−^ in the cultured cells. However, this evidence suggests La^−^ can influence myotube formation, muscle hypertrophy and regeneration, albeit with caution warranted translating these findings to humans.

Kyun et al. (2020) orally administered La^−^ (2 g/kg) in rats to investigate if La^−^ could induce MPS and if responses were time-dependent. Rats were randomly assigned to 4 groups and sacrificed at 0 min (control group), 30 min, 60 min, or 120 min following La^−^ administration. Blood La^−^ was significantly increased to 3.2 mmol/L at 60 min compared to the other times (60 min vs. 0, 30, 120 min; *p* < 0.001), while insulin significantly decreased at 60 min (*p* < 0.001) and 120 min (*p* < 0.01) compared to 0 min. IGF1 mirrored the insulin response, although IGF1 levels were significantly reduced at 30 min (30 min vs. 0; *p* < 0.01) and continued to decrease at 60 min (60 min vs. 30; *p* < 0.01) and 120 min (120 min vs. 30; *p* < 0.05). La^−^ influenced expression levels of genes related to protein synthesis: Akt, mTOR, and IGF receptor. Akt expression was significantly elevated at 30, 60, and 120 min (all *p* < 0.001 vs. 0 min). IGF receptor and mTOR expression were significantly elevated at 60 min relative to baseline (60 min vs. 0 min; *p* < 0.01). La^−^ did not influence expression levels of genes related to protein degradation (i.e., Muscle RING-finger protein-1 [MuRF1] and Muscle-specific F-box protein [atrogin-1]). Phosphorylation is required to activate Akt and mTOR. La^−^ administration did not increase Akt phosphorylation at any time, whereas mTOR phosphorylation significantly increased at 30 min compared to baseline (*p* < 0.05). The data presented in this study agree with previous work [128] suggesting administration of La^−^ in rats can positively influence anabolic signaling via factors related to the Akt/mTOR pathway. However, recent data [140] contradict these findings and will be subsequently discussed.

A study by Shirai et al. (2021) found that intraperitoneally administered La^−^ (1 g/kg of body weight) in calorie-restricted mice for 14 days suppressed atrophy via a direct effect on p70S6K and S6, which are anabolic pathways downstream of mTOR. However, no differences of mTOR phosphorylation levels were detected between the calorically restricted mice with or without La^−^ administration. When comparing the calorically restricted groups, myofiber CSA was significantly greater when mice were administered La^−^. Additionally, La^−^ enhanced mitochondrial enzyme activity and up-regulating proteins associated with aerobic metabolism (i.e., PGC-1α). These findings suggest that La^−^ had a muscle-preserving effect while potentially improving mitochondrial function and aerobic metabolism. An important consideration is La^−^ converts to energy which may influence the total energy intake, thus changing the caloric deficit between the calorie-restricted groups.

Recently, Shirai et al. (2022) have provided data suggesting rats intraperitoneally administered La^−^ does not induce an acute MPS response nor enhance skeletal muscle hypertrophy. Analyses included blood La^−^, plantaris muscle fiber CSA, and WB (assessing for p70S6k, rpS6, pS6, 4EBP1, ERK1/2, and p38). One of the experimental designs explored chronic mechanical overload via synergist ablation (removal of gastrocnemius and soleus) while administering phosphate-buffered saline (PBS) or La^−^ (1 g/kg of body weight) over a 14-day period. Mechanical overload increased the plantaris muscle wet weight (*p* < 0.05) and fiber CSA (*p* < 0.05), however, La^−^ administration did not affect these results. The second experiment explored the acute effects of electrical stimulation (ES) and determined if the addition of La^−^ could enhance acute anabolic signaling or induce MPS. The right gastrocnemius muscle was exercised via ES (100 Hz; 3 s of 10 contractions with 7-s intervals between contractions; 5 sets of 3-min intervals between sets): the left gastrocnemius served as a control. Electrically induced muscle contractions in human muscles (myotubes) has been demonstrated to increase myobundle size and sarcomeric protein abundance in addition to enhanced glycolytic flux (decreased glucose, increased La^−^) and fatty acid oxidation: comparatively greater myotube hypertrophy and upregulated mTORC1 and ERK1/2 activity was observed in 10 Hz versus 1 Hz stimulation [141]. In the current study, ES enhanced MPS and phosphorylation of p70S6K and S6 (*p* < 0.05), however, La^−^ administration did not have an additive effect. La^−^ failed to acutely or chronically affect hypertrophy, nor did it seemingly provide an additive effect to exercise-induced mTORC1 or ERK signaling pathways. One point of speculation is that SA causes massive and rapid hypertrophy due to the supraphysiological response not typically observed in resistance training with humans [142]. Considering this, the protein synthetic response may already be maxed out under this condition, and thus, La^−^ would not have an additive role. On the contrary, no differences in acute intracellular signaling measures were detected following ES, which draws into question the role of La^−^ in skeletal muscle hypertrophy.

The collective data of the discussed studies suggest La^−^ may play a role in skeletal muscle hypertrophy and regeneration through activation of SCs and various anabolic signaling pathways, as well as providing a potential muscle-preserving effect in calorically restricted environments. However, the studies discussed above have conflicting findings, especially regarding the exact mechanism(s) through which La^−^ influences muscle hypertrophy. A summary of the studies suggests that La^−^ may influence various stages of satellite cell proliferation and differentiation, MPS (e.g., mTOR, IGF1, phosphorylation of p70S6K and 4EBP1), and increased Fst (an inhibitor of Mstn). It is important to note that the studies presented above either orally administered La^−^ in mice or cultured C2C12 cells in La^−^ solutions ranging from 2 mmol/L to 20 mmol/L with the higher doses (i.e., ≥10 mmol/L) being levels typically observed in the blood during high-intensity training. Another important point is to consider the time these cells were continuously exposed to a La^−^ treatment, with some studies ranging from 2–6 h while others lasted up to 5 days. This is beyond the threshold that most individuals would exercise, let alone be able to sustain during glycolytically demanding exercise, which complicates practical implications. Differences in unveiling a direct mechanism for La^−^ regulating myogenesis and various anabolic signaling (e.g., mTOR, MAPK/ERK, and GPR81) might be attributed to the cell type used; previous studies have noted this can influence the activation of ERK-related pathways either promoting or inhibiting myogenic differentiation [143]. Cell cultures were obtained from different parties across the in vitro studies. With the previous studies suggesting La^−^ as a potential signaling molecule for hypertrophy, questions arise regarding how these findings translate to humans.

Liegnell et al. (2020) investigated the effects of administering exogenous La^−^ locally during a single-leg knee extension exercise in human subjects. The study investigated intracellular signaling, fractional protein synthesis rate (FSR), and blood/muscle lactate concentrations to determine La^−^ potential role as a signaling molecule in human tissue. Using a randomized crossover design, the training protocol consisted of 6 sets of 8–10 reps at 75% of the subject’s 1RM, whereby subjects received either a sodium La^−^ infusion or a saline solution as a control measure via the antecubital vein. The sodium La^−^ infusion contained 1 mol of La^−^, with an infusion rate of 0.05 mmol/kg/min. Blood La^−^ samples were drawn repeatedly while muscle biopsies were collected at rest, within 2 min of the final contraction, and at 90 min, 3- and 24-h marks post-exercise. No differences in FSR were found between the saline (0.067%/h) or La^−^ infusion (0.062%/hr) with both conditions increasing phosphorylation of mTOR (40–45%), S6K1^T389^ (~3-fold), and p44^T202/T204^ (~80%). Blood La^−^ levels differed significantly by 130% (*p* < 0.001) between the saline trial La^−^ infusion (3.0 vs. 7.0 mmol/L). Muscle La^−^ levels differed between the conditions (*p* < 0.05), with the saline trial measuring 27 mmol/kg dry weight (6.20 mmol/L i.c. water) and the sodium La^−^ infusion measuring at 32 mmol/kg dry weight (7.25 mmol/L i.c. water). Critically, however, La^−^ was intravenously injected in the upper extremity (antecubital vein) during a lower body exercise protocol. Additionally, neither protocol reached the level of La^−^ concentrations within the muscle or blood that the aforementioned studies [110,121,128,129,144] reported anabolic signaling (~10 mmol/L or greater).

The previous in vitro and rodent model studies suggest that La^−^ may contribute to hypertrophy via different mechanisms (Table 1). In contrast, Liegnell et al. (2020) found no additional benefit to exogenous La^−^ and enhanced hypertrophic signaling in human subjects. More research is warranted in humans, considering the intriguing rodent and in vitro work and the paucity of human data on the topic. Questions remain. Under the context that La^−^ may positively influence hypertrophic signaling, do training protocols that elicit the highest levels of La^−^, specifically during resistance training, lead to more hypertrophy? Would emphasizing high La^−^ accumulation (blood/muscle) regularly result in greater hypertrophy outcomes than low muscle and blood La^−^ accumulation? While this is not yet entirely clear, a discussion of studies that have assessed La^−^ in response to resistance training is provided in the next section.

## 4. Section III

### Blood Lactate Response to Various Training Protocols

Variations in blood La^−^ measurements exist in the presented research. This can likely be explained due to different training protocols implemented utilizing both single- and multi-joint exercises of either the upper or lower extremities at different intensities and rest periods. Another important consideration is that studies measured blood La^−^ from either the earlobe, fingertip, or antecubital vein, which can influence results. The time points of blood lactate collection also varied across studies. Some of the highest blood La^−^ measurements were reported when 1-min rest periods were coupled with sets of 10 repetitions at a load that was 15% greater than the subjects 10RM, requiring them to perform ‘forced’ reps during the bench press and half squat [146]. Weakley et al. (2017) [147] also reported some of the highest blood La^−^ values when compounding multiple exercises for both the upper and lower body. Interestingly, the highest blood La^−^ levels were reported when a vibration stimulus was applied to the cable during lat pulldowns for 4 sets to failure at 55% of the subjects’ 1RM [148]. Multiple studies have demonstrated that the mode of contraction and number of repetitions influenced the blood La^−^ response when time under tension was equated between various training protocols [149,150,151]. It is important to note that the studies linking hypertrophy and La^−^ in mice and in vitro were administered with La^−^ levels between 10–20 mmol/L for extended periods. Only a few studies have reported values that exceeded 10 mmol/L (Table 2), but to the authors’ knowledge no study has achieved blood La^−^ values that reached 20 mmol/L during resistance training. Additionally, the data on muscle La^−^ levels during various training protocols is scarce. Future research can help to uncover which training intensities or protocols may elicit the highest blood La^−^ response to reproduce similar conditions from which La^−^ was demonstrated to stimulate hypertrophic signaling [110,121,122,123,129,144].

## 5. Discussion

Energy systems (e.g., glycolysis) supply ATP to support the required energy demands of exercising muscle during resistance training. Hypertrophy-oriented training is reliant on glycolysis, and subsequently, La^−^ production will increase. Some evidence suggests glycolysis may regulate basil mTOR signaling as well as muscle contraction-induced mTOR/4E-BP1 signaling, however, MPS and mTOR/p70SK6 signaling induced via muscle contraction work independent of glycolysis [157]. Some may argue that mechanical tension is the key to stimulating muscle hypertrophy, and that other potential pathways do not influence muscle growth to a meaningful degree or at all. While we do not disagree considering the available evidence, it is important to acknowledge that appreciable mechanical tension for multiple repetitions cannot be completed without the accumulation of various metabolites such as La^−^. The accumulation of metabolites (e.g., La^−^) from resistance training has been termed “metabolic stress” in various studies discussing potential mechanisms for hypertrophy [158,159]. It seems reasonable to hypothesize that mechanical tension-responsive and metabolic stress-responsive pathways influencing muscle hypertrophy can work in tandem. Designing resistance training interventions with the intent to maximize hypertrophy by exploiting both mechanical tension and lactate-induced metabolic stress-related anabolic pathways warrants further research. Willkomm et al. (2014) suggest a sequencing of training starting with an emphasis on maximal eccentric contractions at ≥80% as this has been demonstrated to promote a significant increase in the SC number, followed by a high intensity endurance-based training cycle to induce significant La^−^ levels to initiate early cell cycle withdrawal and early differentiation. The table is provided as a reference for possible training protocols that may be beneficial when programming for a high La^−^ response. Interestingly, Ojasto & Häkkinen (2009) measured blood La^−^ levels ≥ 10 mmol/L when implementing eccentric overload of 70–100% of subject’s 1RM during the eccentric phase of a bench press for 4 sets of 10 repetitions each. This provokes an interesting question: Does combining a maximal La^−^ response with eccentric overload within the same training bout create a synergistic effect or would the high La^−^ response conflict with the signaling produced from the eccentric overload?

Another important point to consider is that there are high intersubject differences in the La^−^ response to the same relative loading condition. Furthermore, within the same individual, lactate responses may vary significantly between various relative loading conditions (e.g., 40% vs. 60% vs. 80% of a 1RM). This also begs the question of whether or not the peak La^−^ value achieved during resistance training is as important as the percent change from baseline values. If La^−^ is indeed a signaling molecule for hypertrophy, and high intersubject differences exist, could the La^−^ response be a determining (or limiting) factor in how well an individual will respond to hypertrophy training? The study by Liegnell et al. (2020) seems to suggest that intravenous infusion of La^−^ does not provide an additional benefit towards anabolic signaling (mTOR and ERK). The authors provide four potential reasons for this: the blood La^−^ levels were insufficient to stimulate a direct signaling response, intramuscular La^−^ levels are key for the stimulation of a signaling response with differences in muscle La^−^ being minimal, anabolic signaling from muscle contractions override the potentially smaller signaling properties of La^−^, or that La^−^ does not possess anabolic signaling in human skeletal muscle. The use of tracers may have provided the ability to detect exactly where the exogenous La^−^ went and if or how much made it to the exercising tissue (e.g., VL). However, a statistically significant increase of ~20% was measured in comparison to the saline control trial which suggests that at least some of the supplemental La^−^ entered the muscle tissue. Another explanation for the lack of an enhanced MPS response might be that receptors through which La^−^ mediates anabolic signaling had become saturated in response to resistance training, and any additional increase in La^−^ provides no additional benefit to an acute response of resistance training. Lastly, endogenously produced La^−^ may result in unique anabolic signaling processes compared to exogenously provided La^−^.

### Future Directions

From a historical perspective, the research and overall scientific view on lactate has been incorrect. For over 200 years it was believed that hLA, not La^−^, was produced during exercise as a waste product resulting in muscle fatigue and soreness; this theory has been debunked. La^−^ should be viewed as a valuable metabolic fuel source that functions as a signaling molecule and epigenetic modifier, with evidence that it may also influence skeletal muscle hypertrophy. A recent review by van Gemert et al. (2022) [160] discussed a potential role for La^−^ as a therapeutic intervention for traumatic brain injury and cardiac failure. Brooks et al. (2021) [29] brilliantly termed La^−^ “a phoenix risen” in the research. Still, it remains unclear as to if and/or how La^−^ influences skeletal muscle hypertrophy. A goal of the current review was to present the cumulative data to make the current research actionable, progress the field’s understanding of La^−^, and highlight areas for future research.

## Figures and Tables

**Figure 1 jfmk-07-00081-f001:**
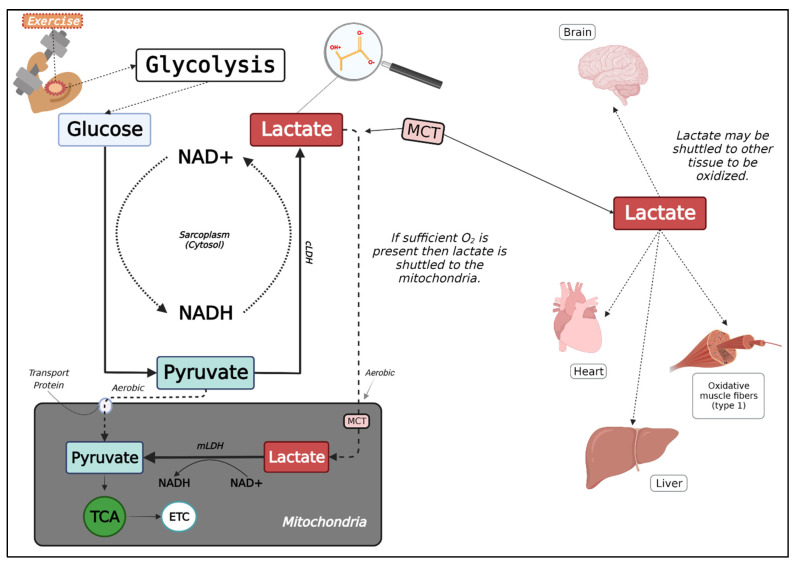
Lactate Metabolism and Shuttle. NAD+/NADH = Nicotinamide adenine dinucleotide (NAD) + hydrogen (H); MCT = monocarboxylate transporter; cLDH = Cytosol lactate dehydrogenase; mLDH = Mitochondrial lactate dehydrogenase; TCA = Tricarboxylic acid cycle; ETC = Electron transport chain. Created with BioRender.com.

**Figure 2 jfmk-07-00081-f002:**
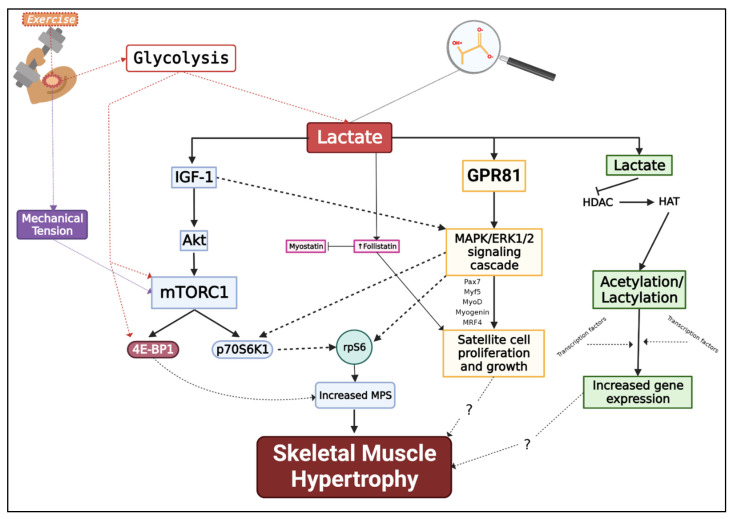
Potential Anabolic Pathways Lactate Influences for Skeletal Muscle Hypertrophy. IGF-1 = Insulin like growth factor 1; Akt = Protein kinase B (PKB); mTORC1 = Mammalian target of rapamycin complex 1; 4E-BP1 = Eukaryotic translation initiation factor 4E (eIF4E)-binding protein 1; p70S6K1 = Ribosomal protein S6 kinase beta-1; GPR81 = G protein-coupled receptor 81; MAPK = Mitogen-activated protein kinase; ERK1/2 = Extracellular signal-regulated kinase 1/2; Pax7 = Paired Box 7; Myf5 = Myogenic factor 5; MyoD = Myoblast determination protein 1; MRF4 = Myogenic regulatory factor 4; HDAC = Histone deacetylases; HAT = Histone acetyltransferases; MPS = Muscle protein synthesis. Created with BioRender.com.

**Table 1 jfmk-07-00081-t001:** Summary of in vitro and in vivo studies investigating anabolic effects of La^−^ administration.

Study	Variable	Results	In Vitro or In Vivo
Wilkomm et al. (2014) [122]	Casp-3	↑ @ 20 mmol @ Days 1 & 3	in vitro
	Ki67	↓ @ 10 & 20 mmol	
	Pax7	↓ on Day 1, → Day 3, ↑ Day 5 @ 10 & 20 mmol	
	Myf5	↓ on Day 1, → Day 3, ↑ Day 5 @ 10 & 20 mmol	
	Myogenin	↑ quickest in control, 10 mm reached same value by Day 5, 20 mmol reached same value at Day 10	
Oishi et al. (2015) [110]	mTOR	↑ @ 10 mmol	in vitro
	p70S6K	↑ @ 10 mmol	
	Fst	↑ @ 10 mmol	
	Mstn	↓ @ 10 mmol	
	Myofibrillar protein content	→	in vivo
	Akt	→	
	Myogenin	↑ (LC; TA/GA)	
Wilkomm et al. (2017) [123]	p38	↓ @ 20 mmol	in vitro
	H3k4me3	↓ @ 20 mmol	
	Myf5	↓ @ 20 mmol	
	Myogenin	↓ @ 20 mmol	
	MHC 1	↓ @ 20 mmol	
	pp38 MAPK	↓ @ 30 min, ↑ @ 1, 4, 24-h. post	in vivo
	H3k4me3	→ or ↓ @ 4 h.	
Tsukomoto et al., 2018 [121]	Myf5	→ @ 10 mmol	in vitro
	Myogenin	→ @ 10 mmol	
	MyoD	↑ @ Days 3 & 5	
	Myf4	↑ @ Days 3 & 5	
	p70S6K	→	
	Myh1	↑ @ ≥ 8 mmol	
	Myh4	↑ @ ≥ 8 mmol	In vivo
	MyoD	→ (trend towards increase)	
	Myogenin	→	
Ohno et al. (2018) [144]	GPR81	↑ expression @ mRNA & protein levels in myotubes and myoblasts with the highest levels in myotubes.	in vitro
	Myotube diameter	↑ @ 20 mmol	
	MEK1/2	↑ @ 20 mmol	
	p-ERK1/2	↑ @ 20 mmol	
	p-p90RSK	↑ @ 20 mmol	
	p-Akt	→ @ 20 mmol	
	p-mTOR	→ @ 20 mmol	
	p-p70S6K	→ @ 20 mmol	
	p-FoxO3a	→ @ 20 mmol	
	p-ULKI	→ @ 20 mmol	
	LC3B-II	→ @ 20 mmol	
Cerda-Kohler (2018) [128]	Lactate	↑ in lactate vs. control	in vivo
	Blood glucose	→ in lactate vs. control	
	Insulin	→ in lactate vs. control	
	ERK1/2	↑ @ 40 min in lactate for quadricep, → EDL or Soleus	
	IGF-1	→	
	Akt	↑ @ 40 min in lactate for quadricep & EDL	
	p70S6K	↑ in quadricep only	
	S6	↑ for quadricep & EDL, → Soleus	
	AMPK	→ in quadriceps or EDL, ↑ in Soleus	
	ACC phosphorylation	↑ in Soleus	
	TBC1D1	→ for all muscle groups	
	TBC1D4	↑ for Soleus, → quadriceps & EDL	
	PDH-E1α	↓ in Soleus, → or ↓ in EDL, → or ↑ in EDL	
Ohno et al. (2019) [109]	Bodyweight	→ (oral administration)	in vivo
	TA muscle weight	↑ (oral administration)	
	Fiber CSA	↑ (oral administration)	
	Pax7-positive nuclei	↑ 200% and 138% at Weeks 1 & 2 (oral administration)	
	Bodyweight (CTX)	→ CTX-injected groups	
	TA muscle weight (CTX)	↑ in LX (24% absolute & 29% relative)	
	Fiber CSA (CTX)	↑ 44% at 2 weeks in LX vs. 1 week	
	Pax7-positive nuclei (CTX)	↑ (0.08/myofiber) vs. control (0.02/myofiber)	
		↑ in LX vs. CX	
	Myotube	↑ in diameter, length, and myonuclei @ 20 mmol for 5 days	in vitro
Kyun et al. 2020 [129]	Blood lactate	↑	in vivo
	Insulin	↓	
	IGF1	↓	
	Akt mRNA	↑	
	mTOR mRNA	↑	
	IGF receptor mRNA	↑	
	phosphorylation of Akt	↑	
	phosphorylation of mTOR	↑	
	Atrogin-1	→	
	MuRF1	→	
Shirai et al., 2021 [145]	Myofiber CSA	↓ vs. control, ↑ with CR + lactate vs. CR alone	in vivo
	Bodyweight	↓ vs. control, → with CR + lactate vs. a ↓ CR alone	
	Food intake	↓ vs. control, same between both CR groups.	
	Akt	↓ vs. control	
	mTOR	↓ vs. control	
	p70S6K	↓ vs. control, ↑ vs. CR.	
	4EBP1	↓ vs. control	
	S6	↓ vs. control, ↑ vs. CR.	
	MAFbx	→ between groups	
	MuRF1	→ between groups	
	LC3-II/LC3-I	↑ in CR, but not in CR + Lactate	
	p62	↓ in CR, but not in CR + Lactate	
	Ubiquitinated protein level	↓ in CR, but not in CR + Lactate	
	AMPK	→ between groups	
	PGC-1α	↑ in CR + Lactate & CR	
	UQCRC2	↑ in CR + Lactate & CR	
	MTCO1	↑ in CR + Lactate & CR	
	ATP5A	↑ in CR + Lactate & CR	
	NDUB	↑ in CR + Lactate & CR	
	SDHB	↑ in CR + Lactate & CR	
	CS activity	↑ in CR + Lactate vs. CR	
	Enzyme activity	↑ in CR + Lactate vs. CR	
Shirai et al., 2022 [140]	Plantaris weight	→ between Lactate-OL and PBS-OL	in vivo
	Plantaris CSA	→ between Lactate-OL and PBS-OL	
	p-p70S6K (Thr389)	→ between Lactate-OL and PBS-OL	
	total-p70S6K	→ between Lactate-OL and PBS-OL	
	p-S6	→ between Lactate-OL and PBS-OL	
	total-S6	→ between Lactate-OL and PBS-OL	
	p-4EBP1 (Thr37/46)	→ between Lactate-OL and PBS-OL	
	total-4EBP1	→ between Lactate-OL and PBS-OL	
	p-p70S6K (Thr389)	→ between Lactate-ES and PBS-ES	
	total-p70S6K	→ between Lactate-ES and PBS-ES	
	p-S6	→ between Lactate-ES and PBS-ES	
	total-S6	→ between Lactate-ES and PBS-ES	
	p-4EBP1 (Thr37/46)	→ between Lactate-ES and PBS-ES	
	total-4EBP1	→ between Lactate-ES and PBS-ES	
	p-ERK1/2 (Thr202/Try204)	→ between Lactate-ES and PBS-ES	
	total-ERK1/2	→ between Lactate-ES and PBS-ES	
	p-p38 (Thr180/Try182)	→ between Lactate-ES and PBS-ES	
	total-p38	→ between Lactate-ES and PBS-ES	
	Puromycin	→ between Lactate-ES and PBS-ES	

Key. ↑ = increase; → = no change/difference; ↓ = decreased.

**Table 2 jfmk-07-00081-t002:** Blood Lactate Responses to Various Resistance Training Protocols.

Study	Training Protocol	Bla- (mmol/L)	Measurement Site	Exercise(s)
Haddock & Wilkin (2006) [152]	3 sets of nine exercises @ 8 RM to volitional fatigue	10.2	Fingertip	Bench press, lateral pull down, leg curl, overhead press, knee extension, biceps curl, triceps pull down, and abdominal crunch.
Boroujerdi & Rahimi et al. (2008) [146]	4 sets × 10RM + 15% additional load with 1-min rest period * 3–4 forced reps were completed for both protocols once subjects could not complete a rep on their own.	14.5	Antecubital vein	Bench Press and Half Squat
Ojasto & Häkkinen (2009) [153]	4 sets × 10 repetitions @ 70% of 1RM for the eccentric phase and 70% of 1RM for concentric phase with 2-min rest period	10.23	NA	Bench Press
	4 sets × 10 repetitions @ 80% of 1RM for the eccentric phase and 70% of 1RM for concentric phase with 2-min rest period	10.99	NA	Bench Press
	4 sets × 10 repetitions @ 90% of 1RM for the eccentric phase and 70% of 1RM for concentric phase with 2-min rest period	12.12	NA	Bench Press
	4 sets × 10 repetitions @ 100% of 1RM for the eccentric phase and 70% of 1RM for concentric phase with 2-min rest period	11.77	NA	Bench Press
Kelleher et al. (2010) [154]	4 supersets × 10 reps @ 70% to volitional failure with 60-s rest periods. * agonist-antagonist supersets	10.79	Fingertip	Bench Press, bent-over row, biceps curl, lying triceps extension, leg extension, and leg curl
Sánchez-Medina & González-Badillo (2011) [155]	3 sets × 10 reps @ RIR 2 with 5-min rest periods	10.6	Fingertip	Back Squat
	3 sets × 12 reps @ RIR 0 with 5-min rest periods	12.4	Fingertip	Back Squat
	3 sets × 10 reps @ RIR 0 with 5-min rest periods	11.7	Fingertip	Back Squat
	3 sets × 8 reps @ RIR 0 with 5-min rest periods	10.4	Fingertip	Back Squat
	3 sets × 6 reps @ RIR 0 with 5-min rest periods	10	Fingertip	Back Squat
Paoli et al. (2012) [156]	2–3 sets @ 6RM w/2–3 additional reps with 20-s rest period between exercises and 2:30 min between rounds.	10.5	Earlobe	Leg press, bench press, dorsal machine
Couto et al. (2013) [148]	4 sets to failure @ 55% of 1RM with 2-min rest periods	14.76	Ulnar vein	Lat Pulldown
	4 sets to failure @ 55% of 1RM with 2-min rest periods * Local vibration applied to cable	16.92	Ulnar vein	Lat Pulldown
Gorostiaga et al. (2014) [48]	5 sets × 10 reps @ 10RM with 2-min rest periods	10.3	Earlobe	Leg Press

## Data Availability

Not applicable.

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
