# Peer review of "Beyond Mechanical Tension: A Review of Resistance Exercise-Induced Lactate Responses & Muscle Hypertrophy"

_jfmk, 2022, doi:10.3390/jfmk7040081_

Round 1

Reviewer 1 Report

General comment and recommendations

This review article discusses an interesting topic, namely the potential role of lactate in muscle hypertrophy. Over the past decade, more and more animal studies, performed either in in vitro or in vivo conditions, have indicated that lactate may play a role in muscle hypertrophy or in the prevention of muscle atrophy. A few human studies have also provided indirect support for a potential role of lactate as an anabolic agent for increased muscle mass following a resistance training program.

This review is timely and includes the vast majority of articles published to date in the field. However, even though the paper is relatively well done, some parts are hard to follow mainly due to the lack of figures associated with the text. In different sections, the introduction of diagrams would make the article easier to follow , notably by a non-specialist readership. As an example, the main pathway(s) involved and the processes by which lactate might promote a hypertrophic response in muscle tissue would be very helpful (section II).

Although the reviewer acknowledges that there are more studies carried out in animals than in humans, the content of the review is quite disproportionate especially with far more details reported for the animal studies compared to the human studies. In addition, the introductory part and the first section dealing with lactate in general could be shortened because the main objective (and originality) of the paper is not to review the mechanisms of lactate production but rather its potential relationship to muscle hypertrophy. As an example, the difference in how lactate is measured (blood vs muscle) can be summarized. On the other hand, the reason why high lactate production during training using anaerobic exercises (e.g. training for 400m race) does not lead to a high level of muscle hypertrophy could be evoked and discussed somewhere in the paper.

P. 2, para 5, l. 11 : “red muscle”: it is an old terminology. I suggest changing it by something like : “…within muscles composed by slow-oxidative fibers (type I) during fully aerobic conditions…”.
P. 5 (The lactate shuttle hypothesis): a figure might be useful to illustrate this section.
P. 10, para 2, l. 14: “experiment” and “myogenesis” are not spelled correctly.
P. 11 para 3, l. 25-31:  is there any evidence that high blood lactate concentration could be attained when different exercises involving different large muscles groups are combined in a same session?
P. 12, para 3, l. 16-17 and last sentence: these sentences are incomplete.
P. 13, para 2, l. 7-9: more details are needed regarding the protocol and the stimulation characteristics as they can influence the metabolic responses.
P. 16, para 1, l. 24-29…: these results are a little surprising because motor unit activity during an eccentric contraction is known to be lower (for similar absolute force value) or ~equal (for similar relative force value) than during a concentric contraction.
P. 16, last para : the duration of the rest period between sets also influences the lactate response. In that context, this information is lacking for two references in Table 1.

Author Response

Thank you very much for your review. We have responded to each of your comments on a point-by-point basis and made corresponding revisions to the manuscript (highlighted in red). We hope that the revisions meet with your satisfaction.

Reviewer 2 Report

General Comments: Certainly an impressive and thorough review of lactate and a well-thought outline of the history, measurement, and implications of muscle hypertrophy.  At times, this manuscript became very granular, and I feel a small audience will only understand it, but considering the topic, this is unavoidable.  I know the co-authors of this manuscript have a well-documented background in muscle hypertrophy and anabolic factors which contribute to hypertrophy.  The table on lactate and resistance training was well-done and very helpful to gap the practical applications of the type of programs that induce the most lactate, even though it appears that lactate is not much of a driver for muscle hypertrophy. 

I don't have any significant suggestions.  I have some minor comments with the references, as I don't see some references in the bibliography.  Considering there are over 150 references, I could see this mistake happening.  I would advise the author to carefully review the references in the manuscript again and ensure all are in the bibliography. 

SECTION II: Minor change: On page 7, the author states…" following resistance training have not been demonstrated to meaningfully influence muscle hypertrophy (Wilkinson et al., 2006; Spiering et al., 2008; West et al., 2009)." Can the authors provide more recent studies about the "hormone hypothesis"?

FUTURE DIRECTIONS: Minor change: Page 17, "the author states...Liegnall et al. (2020) seems to suggest that intravenous infusion of La- does not provide an additional benefit towards anabolic signalling (mTOR and ERK)." I can't see to find this study on the references page.

FUTURE DIRECTIONS: Minor change: On page 17, The author states…" recent review by Germert et al. (2022) discussed a potential role for La- as a therapeutic intervention for traumatic brain injury and cardiac failure".  Again, I can't see to find this study on the references page either.

Author Response

(The authors gave the same response as above.)
